# Inter-Specimen Analysis of Diverse Finite Element Models of the Lumbar Spine

**DOI:** 10.3390/bioengineering11010024

**Published:** 2023-12-26

**Authors:** James Doulgeris, Maohua Lin, William Lee, Kamran Aghayev, Ioannis Dimitri Papanastassiou, Chi-Tay Tsai, Frank D. Vrionis

**Affiliations:** 1Department of Medical Engineering, University of South Florida, Tampa, FL 33620, USA; jdoulger@usf.edu (J.D.); wlee2@usf.edu (W.L.); 2Department of Ocean & Mechanical Engineering, Florida Atlantic University, Boca Raton, FL 33431, USA; tsaict@fau.edu; 3Department of Neurosurgery, Esencan Hospital, Baglarcesme Mahallesi, Istanbul 34510, Turkey; kamranag@gmail.com; 4Department of Orthopedic, General Oncological Hospital Kifisias “Agioi Anargryroi”, 14564 Athens, Greece; jpapa73@yahoo.gr; 5Department of Neurosurgery, Marcus Neuroscience Institute, Boca Raton Regional Hospital, Boca Raton, FL 33486, USA

**Keywords:** lumbar spine, nonlinear finite element model, biomechanical analysis, von-Mises Stress, ROM

## Abstract

Over the past few decades, there has been a growing popularity in utilizing finite element analysis to study the spine. However, most current studies tend to use one specimen for their models. This research aimed to validate multiple finite element models by comparing them with data from in vivo experiments and other existing finite element studies. Additionally, this study sought to analyze the data based on the gender and age of the specimens. For this study, eight lumbar spine (L2–L5) finite element models were developed. These models were then subjected to finite element analysis to simulate the six fundamental motions. CT scans were obtained from a total of eight individuals, four males and four females, ranging in age from forty-four (44) to seventy-three (73) years old. The CT scans were preprocessed and used to construct finite element models that accurately emulated the motions of flexion, extension, lateral bending, and axial rotation. Preloads and moments were applied to the models to replicate physiological loading conditions. This study focused on analyzing various parameters such as vertebral rotation, facet forces, and intradiscal pressure in all loading directions. The obtained data were then compared with the results of other finite element analyses and in vivo experimental measurements found in the existing literature to ensure their validity. This study successfully validated the intervertebral rotation, intradiscal pressure, and facet force results by comparing them with previous research findings. Notably, this study concluded that gender did not have a significant impact on the results. However, the results did highlight the importance of age as a critical variable when modeling the lumbar spine.

## 1. Introduction

Lumbar spine morbidities are highly prevalent and affect a significant portion of the population [1]. These conditions can result in various symptoms, ranging from lower back pain to spinal deformities. Even mild lower back pain can significantly impact the quality of life for patients, leading them to seek remedies and prompting surgeons to study current standards of care. However, extensive research is required to ensure that treatments offer more benefits than drawbacks, given the complex biomechanics of the lumbar spine [2]. Consequently, the investigation of medical devices and treatments in this field has gained substantial popularity in the literature [3]. These studies primarily focus on in vitro (cadaver studies) [4,5,6,7] or finite element analysis (FEA) methodologies [8,9,10,11].

FEA has emerged as a favored approach for studying the biomechanics of the lumbar spine [12,13,14,15,16] and the cervical spine [17,18,19,20]. FEA enables the capture of valuable data, such as contact forces, strains, stresses, micromotion, and pressures, which are challenging or even impossible to accurately obtain through in vivo or in vitro studies. The cost-effectiveness, speed, and repeatability of FEA make it an attractive choice, especially for comparative analyses, when compared to in vitro studies. While biomechanical FEA of the spine is currently an accepted research practice, several challenges exist in accurately modeling the spine [21,22,23]. Loading methodologies vary among researchers in both in vitro [24] and finite element studies of the lumbar spine [14,15,25,26]. The complex loading of the spine [27] may contribute to these non-standardizations, but a study in the literature compares various loading methodologies and provides recommendations for FEA [28]. The anatomical complexity and inter-subject variability of vertebral structures make it difficult to compare findings with other studies, necessitating the inclusion of multiple models [29]. However, trends in methodology have helped mitigate the issue of the loading methodology in computational studies, and each study contributes to enhancing the predictive analysis of FEA.

Several authors have conducted FEA on the lumbar spine, with some focusing on functional spinal units [14,30], while others concentrate on multi-segmental analyses involving at least three lumbar levels. Xu et al. developed five models covering the L1–L5 region based on healthy living subjects, applying 7.5 Nm of torque with varying compressive force methodologies [25]. Their study involved a convergence analysis, Von Mises stress comparison, and validation against experimental and computational literature. Eiberlein et al. created an L2–S1 model derived from a cadaveric model, subjecting it to 10.0 Nm of torque without a compressive preload [12]. The computational model was compared with experimental data obtained from the cadaver, and a convergence analysis was performed, accounting for disc degeneration. Zhang et al. initially developed an L1–L5 model and later conducted a comparative analysis of surgical implants using an L2–L5 model, subjected to a 280 N compressive load and 7.5 Nm of torque [13]. Dreisharf et al. conducted eight studies involving an L1–L5 model, analyzing pure bending at 7.5 Nm, pure compression up to 1000 N, and combined bending/compressive loading [15].

While boundary conditions vary, the modeling process follows standardized procedures. Researchers typically collect computerized tomography (CT) images and employ various software to construct 3D geometrical surfaces of the bones [13,14,25,26,31]. The base model is then transferred to FEA software, where material properties and loading boundary conditions are incorporated. Although there is some variability in the material boundary conditions among FEA studies [13,14,15,25,26,31], the differences are minor, as evidenced by comparing multiple works [15]. Previous FEA studies have suggested using larger sample sizes to account for inter-specimen variability; however, this issue has not been adequately addressed in FEA studies. FEA analysis of the lumbar spine serves different purposes. In our earlier research, we achieved the successful creation of a sophisticated 3D finite element analysis (FEA) model for the lumbar spine, thereby verifying the precise depiction of ligaments, nucleus pulposus, and annulus using solid modeling techniques [11]. The objective of this study is to validate FEMs (finite element models) of the lumbar spine and investigate the influence of age and gender in these models. To facilitate future comparative analyses between medical devices, the model will incorporate a level above and below the target level of L3–L4.

## 2. Materials and Methods

### 2.1. Model and Material

The researchers obtained eight lumbar and sacral CT scans from patients aged 44–73 years old. The scans had 0.75 mm thick slices. The subjects included four females (aged 52, 58, 62, and 70 years) and four males (aged 44, 54, 62, and 73 years). The authors imported the scans to an anatomical analysis software, Materialise Mimics Innovation Suite (Materialise, Plymouth, MI, USA), to create 3D surface models of the lumbar vertebral bodies (L2–L5) from the 2D CT scans. The software smoothed the shells and eliminated any abnormal bony structures. Additionally, the software generated geometric solids representing the cortical shell and cancellous core. The average thickness of the cortical shell in the anterior region was measured to be 0.75 ± 0.125 mm.

To further process and simulate the soft tissues, the authors transferred the 3D surface models into the 3D parametric design software Solidworks version 2019 (Dassault Systèmes, Velizy-Villacoublay, France). The posterior elements of each vertebra were separated to create solid structures. Cartilaginous bodies with a thickness of 0.75 mm were added to mimic the endplates of the spine. A nucleus pulposus was included at the desired levels, and four concentric rings with a thickness of 1.5 mm were created around each nucleus to form the annulus. Cubical solid bodies were inserted in the center of all the anterior vertebral bodies to act as preload connections.

The models were imported into a finite element analysis software, ANSYS Workbench R19.2 (ANSYS, Canonsburg, PA, USA) for pre-processing. The material boundary conditions used in this study are summarized in Table 1, which was derived from previous literature [12,25,32,33]. Nonlinear tension-only spring elements representing various ligaments (ALL, PLL, TL, FLA, ISL, SSL, and FCL) were added to each spinal level. The stress–strain values for these ligaments were derived from Eberlein et al. (2004) and are provided in Table 2 [12]. Reinforcement tension-only fibers were embedded in the annulus ring bodies at the 25% and 75% locations from the outermost surface of each annulus ring, alternating in ±30° directions. The material properties of these fibers were obtained from the works of Shirazi-Adl et al. The fiber strength decreased as it moved from the outermost layer to the innermost layer, as shown in Table 3 [33]. Meshing consisted of tetrahedral patch-independent body sizing for all bodies except the annulus rings, which used quadrilateral multizone methodology and mapped face meshing. The latter was required to ensure that the fiber reinforcing coding was properly added to the model in the correct directions and locations. All fiber reinforcing was verified through vector images and manual review before solving. Figure 1 illustrates each pre-process model before solving.

### 2.2. Boundary, Contact and Loading Conditions

The inferior surface of the L5 endplate was constrained in all six (6) degrees of freedom. The superior surface of the L2 endplate was unconstrained but used as the location for moment application, shown in Figure 2. A series of linear springs, that cannot exceed a pre-determined max load, and joint bodies, which are connected to the inner surface of the cortical shell via a joint command, are attached to the upper surface of the L2 endplate to simulate the loads from muscles and torso weight. The bottom-most spring in the series is the softest and will reach max preload, and then stay at that load regardless of increased deformation. This replicates the hybrid follower load of the lumbar spine.

The moment and follower preload values were contingent on the intended motion and were as follows: flexion, 7.5 Nm and 1175 N; extension, 7.5 Nm and 500 N; lateral bending, 7.8 Nm and 700 N; and axial rotation, 5.5 Nm and 720 N. These loading conditions were selected from the works of Dreischarf [15] and Rohlmann [28], but the preload methodology was based on Rohlmann’s analysis of the loading methodology [28]. The loading paradigm consisted of 2 steps: step 1 ramped the preload to a maximum of more than 60 s, and step 2 ramped the applied moment maximum of over 120 s. The connections between soft and hard tissues were established using the bonded contact method with symmetrical detection calculations. The frictional contact method (µ = 0.05) was employed for the facet joints of the posterior elements, assuming initial contact within a pinball region of 0.75 mm.

### 2.3. Statistical Analysis

The analysis included statistical analyses with a significance level of 95% (α = 0.05), when the data were available. This began with a one-way ANOVA to determine if there were differences among the groups. If differences were determined, post hoc testing began with a Shapiro–Wilk test to determine normality. If normal, the analysis would precede to comparative parametric testing (unpaired *t*-tests), otherwise, non-parametric testing (U test) was used. If no significant difference was determined between the groups, the analysis considered one-tail testing, where a difference of ±20% of the total mean was considered for equivalence testing.

## 3. Results

While several FEAs have used the L2–L5 section of the lumbar spine, none to the authors’ knowledge have used the loading methodology in this study. The models this study uses for comparison utilize the L1–L5 section, but consistency in the loading paradigm was deemed more viable than the same section of the lumbar spine. The results were compared to in vivo experimental measurements used in the literature [34,35,36] and FEA studies with similar loading paradigms [15,25].

### 3.1. Convergence

The accepted meshes, not including fiber and ligament elements/nodes, resulted in an element range of 698,473–1,064,819 and a node range of 339,695–584,930. The specimen with the least elements/nodes was selected for a convergence analysis, where the mesh was increased by approximately 30% and then solved again. The stress, range of motion, facet forces, and intradiscal pressure were compared against each other. The resulting values did not change by more than 5%; therefore, all models were concluded to be converged.

### 3.2. Intervertebral Rotation

The resulting range of motion or intervertebral rotation is shown in Figure 3 and was calculated by using the average Euler angle of the entire vertebral body. The results are within the range of the other FEA models and are within the in vivo range for all motions excluding flexion. This study was significantly different from the in vivo results for all levels in flexion rotations and the L3–L4 extension rotation. The model had consistently lower median values in flexion and, similar to the FEA literature, was out of range of the in vivo values in some spinal levels. The median values for extension were consistent in all the spinal levels, but the top range was often higher than the FEA models. The lateral bending and axial rotation results were in the range of the in vivo and literature results.

### 3.3. Pressure

This study also considered the intradiscal pressure in the nucleus pulpous. This study measured these results based on the average normal stresses in the axial direction of the nucleus. The in vivo results from the Wilkes study [36] were included, but since it only contained the measurements from one subject in the L4–L5 level, this study utilized the flexion and extension measurements from Sato et al. [35]. For further comparison, this study also included data from other FEA studies [15,25]. A comparison of the pressures can be seen in Figure 4. The results of the intradiscal pressures were statistically similar to the in vivo results in flexion (±0.25 MPa) and extension (±0.15 MPa). This study produced higher median intradiscal pressures in lateral bending and axial rotation compared to the other FEA models but stayed within the range of the results. Only one L4–L5 in vivo measurement was available for lateral bending and axial rotation; therefore, it is uncertain where the acceptable range is for these results.

### 3.4. Facet Forces

The facet forces were also considered in this study. In vivo measurements are not present in the literature; therefore, this study compared the results to the other FEA models [15,25]. These can be seen in Figure 5. The flexion data were omitted to match the data in the literature. The extension median facet forces are larger than what is present in the other FEA models. The lateral bending median facet forces are higher in the L2–L3 and L3–L4 but are consistent in the L4–L5 levels. The axial rotation median facet forces are lower in the L2–L3 and L3–L4 regions but are consistent in the L4–L5 levels. While the median lateral bending and axial rotation values differ from the literature, the values are within the expected range.

### 3.5. Model Validation

The metrics reported in this study produce overlapping results when compared to other validated FEA models in the literature. However, the flexion results of this and other FEA models are not within range of the in vivo results, which indicates that more studies should be included for in vivo validation and/or the model boundary conditions should be investigated further. The model is valid in all other motions for intervertebral rotation. The model is also valid in the pressure results compared to the other FEA studies, but validation can only be claimed for L4–L5 in flexion and extension due to the limited in vivo data for lateral bending and axial rotation. Facet forces were presented, but no in vivo data are available for validation. This model is acceptable for comparative analyses but needs to adjust the boundary conditions for full validation.

### 3.6. Age and Gender Comparison

The authors also compared the age groups and gender results of this study. The gender analysis results produced no significant differences in any of the metrics shown in Figure 6, Figure 7 and Figure 8. The age analysis separated the specimens into two groups (40–59 and 60–75 years old), and the analysis is shown in Figure 9, Figure 10 and Figure 11. There were multiple significant differences between the groups in the following comparisons: all lateral bending intervertebral rotations, L4–L5 axial rotation intervertebral rotation, L3–L4 lateral bending pressure, L4–L5 axial rotation pressure, L2–L3 lateral bending facet force, L4–L5 lateral bending facet force, L2–L3 axial facet force, L3–L4 axial rotation facet force, and L4–L5 facet force.

## 4. Discussion

In this study, the models chosen for validation were from Dreshaf [15] and Xu [25], but these models included the L1–L5 levels, whereas this study utilized the L2–L5 spinal models. While other models have used the L2–L5 spinal levels [37,38], their loading methodology for preload was significantly lower than the preload used in this study. A similar loading methodology is a more critical factor than the same spinal levels, since the analysis is on individual levels and not global motions. The decision not to use the L1–L5 level was based on the desire for future studies to be able to simulate medical device interventions on the L3–L4 level while still having a symmetric level above and below the intervention.

This study utilized spring elements for ligaments and preload application instead of discrete beams as seen in other studies [15,26,28]. There is no fundamental difference between the two, and these will perform similarly in a simulation. However, the use of shell elements may produce differing results, since contact can be included with such elements. One study has performed this inclusion [25], and one of the models from Dreischarf’s study used this methodology [15], but the analysis included these studies for validation and found no range differences between the models. Overall, spring elements are suitable for the lumbar spine finite element model to save on computation time.

The range of motion in the lumbar spine is crucial for functional mobility, spinal health, and preventing lower back pain. It allows us to perform daily activities, supports spinal stability, and reduces the risk of injury. Maintaining a healthy range of motion promotes joint and muscle health, while restricted motion can lead to imbalances and increased stress on spinal structures. This model predicted lower flexion results than what was reported in the in vivo study, but it is within range of the other validated FEA models. Increasing the applied torque alone in flexion may not be sufficient to increase the intervertebral motion to what is reported in the in vivo measurements and may push the intradiscal pressure beyond the acceptable range. We postulate that the ligaments also contribute to this issue. We recommend that future studies investigate a reduced cross-sectional area/change in the nonlinear curve of the ligaments, excluding the anterior and transverse ligaments, to determine whether this is sufficient.

The intradiscal pressure results were consistent with the FEA studies in the literature. The lateral bending and axial rotation results were contained in the top half of the FEA studies and did not capture the single in vivo measurement, but the studies do not explicitly describe how these pressure measurements were obtained. The pressure measurements in this study measure an average pressure in the relative normal plane, which could potentially produce higher results than a single pressure reading in the center of the disc. Since only one pressure reading is available in the in vivo studies for lateral bending and axial rotation, no suggestions or conclusions are possible until more data are available.

The intradiscal pressure results were consistent with the FEA studies in the literature. The lateral bending and axial rotation results were contained in the top half of the FEA studies and did not capture the single in vivo measurement, but the studies do not conclusively indicate how these pressure measurements are obtained. The pressure measurements in this study measure an average pressure in the relative normal plane, which could potentially produce higher results than a single pressure reading in the center of the disc. Since only one pressure reading is available in the in vivo lateral bending and axial rotation study, no suggestions or conclusions are possible until more data are available.

The facet forces are crucial in maintaining the balance between stability and flexibility in the lumbar spine. Excessive or asymmetric facet loading can lead to facet joint degeneration, facet arthropathy, and other spinal pathologies. On the other hand, inadequate facet loading may result in reduced spinal stability and increased stress on other spinal structures. Facet forces play a crucial role in maintaining the balance between stability and flexibility in the lumbar spine. Studying facet forces with FEA enables a better understanding of the factors contributing to facet joint degeneration and can inform interventions and treatments for facet-related disorders. Researchers and clinicians can gain a better understanding of the factors that contribute to facet joint degeneration, such as abnormal spinal alignment, disc degeneration, or changes in muscle forces, by studying this measurement. This knowledge can aid in the development of interventions and treatments that target facet-related disorders. The facet force results were presented in this study, but no conclusions can be drawn in the comparative analysis due to the unavailable data from the literature. Given how important the facets are in limiting extension moments, we determined these results to be acceptable, since the intra-discal pressure was consistent with the literature and there were no in vivo measurements available for comparison.

Studies utilize many different material boundary conditions for soft and hard tissues. This can range from linear approaches to the ligaments and annular fibers to anisotropic cortical bone moduli. Nonlinear values are postulated to be more accurate when extreme forces are used [12,25] but until an extensive material sensitivity analysis is performed, the inclusion of complex material boundary conditions remains ambiguous. The loading paradigm is another boundary condition that can vary in the literature. Rohlman et al. conducted a study wherein they compared six (6) different loading paradigms [28]. Without the application of a preload, intradiscal pressure can produce lower results than what is measured in vivo [28]. Therefore, the selection hybrid model is preferable to achieve both realistic intervertebral motions and intradiscal pressure. Our results support that the loading paradigm is sufficient for all motions except flexion, where other studies have postulated that a larger moment may be beneficial [15].

Adjacent degeneration is a common adverse event to discs which are adjacent to fused segments. The current clinical rationale for this malady is that it occurs due to increased pressure due to increased stiffness in an adjacent level. This model is set up to investigate this issue in the future by considering multiple segments and pressure measurements. However, the in vivo and FEA data are based on quasi-static loading, which only considers the pressure at the end of the load. While fundamentally changing the stiffness of one level may cause an increase in the static adjacent intradiscal pressure, the pressure results in dynamic loading may be more dramatic if the angular rotation remains the same.

This model and the majority of lumbar finite element models use a static loading pattern [15,25,28,37,38,39]. This study used a static loading pattern with a generous loading time to aid in the convergence of the analysis and avoid noise from rapid loading. However, this is different from the actual viscoelastic nature of the intradiscal soft tissues [40,41,42]. The development of these material properties and a dynamic loading paradigm would be beneficial to the literature.

The boundary conditions in FEA models are applied to specimens with no regard for specimen weight, age, bone density, or gender. There are large anatomical differences between people [21,22,23,39,43]. Few studies have included more than a total of four specimens in their data set [25,39], but the literature consistently recommends the use of multiple specimens [15,39]. The results indicate that specimen gender is not sensitive to the boundary conditions utilized in this investigation, but the age of the specimen may present some differences in lateral bending and axial rotations. The boundary conditions are applied consistently among the specimens, which may indicate that there are slight differences in the base structures with respect to the age of the specimens. This also suggests that the inclusion of only one specimen in an FEA study may not predict reasonable data if the boundary conditions are applied with no regard to the specimen metrics. Thus, this study suggests that multiple specimens should be utilized with the presented methodology.

This study’s clinical relevance lies in its advancement of finite element analysis for spinal research. It addressed the common practice of using small-scale models by validating multiple finite element models with in vivo data. Focusing on the lumbar spine, it accurately simulated key motions and analyzed parameters like vertebral rotation, facet forces, and intradiscal pressure. This study’s findings enhance our understanding of spinal biomechanics, with implications for diagnosis and treatment. Notably, age, rather than gender, was identified as a critical variable in modeling the lumbar spine, underscoring its importance in clinical practice.

This study has several limitations that should be acknowledged: it lacks a complete clinical evaluation, omitting crucial information about patients’ underlying diseases, surgical history, or spine conditions that could significantly affect the finite element analysis (FEA) results. Additionally, while age-related variations in bone density may impact FEA outcomes, this study did not thoroughly explore this aspect. The small sample size of eight patients limits the generalizability of the findings, and the absence of detailed patient-specific information, such as general health and lifestyle, further restricts the study’s comprehensiveness. Future research should address these limitations by conducting a more thorough clinical assessment, considering age-related bone density changes, employing larger and more diverse samples, and including specific patient characteristics to enhance the validity and applicability of the results.

## 5. Conclusions

The following conclusions were determined from this study:The methodology produces reasonable representations of lumbar spine biomechanics and is suitable for comparative analyses.Future studies should investigate material and loading conditions.Spring elements are viable additions to lumbar spine finite element modeling.Multiple specimens are needed in lumbar spine finite element modeling to account for specimen variability.The age of the specimens had more noticeable differences compared to the gender in this study.

## Figures and Tables

**Figure 1 bioengineering-11-00024-f001:**
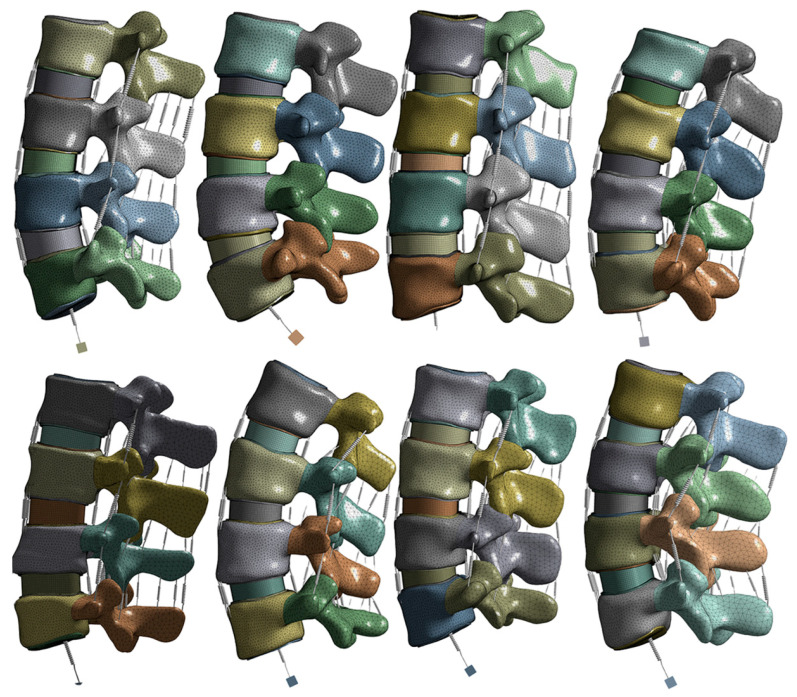
Preprocessed models for the L2–L5 lumbar spines used in this model. (Top left to right: females ages 52, 58, 62, and 70 years old; bottom left to right: males ages 44, 54, 62, and 73 years old).

**Figure 2 bioengineering-11-00024-f002:**
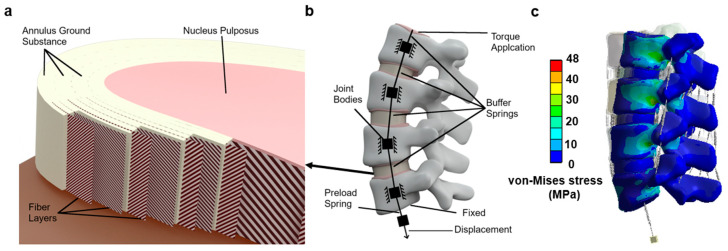
(**a**) Intervertebral disc section view to show the fiber layers; (**b**) example of the loading setup; (**c**) von-Mises stress (MPa) under extension situation with 62-year-old female model.

**Figure 3 bioengineering-11-00024-f003:**
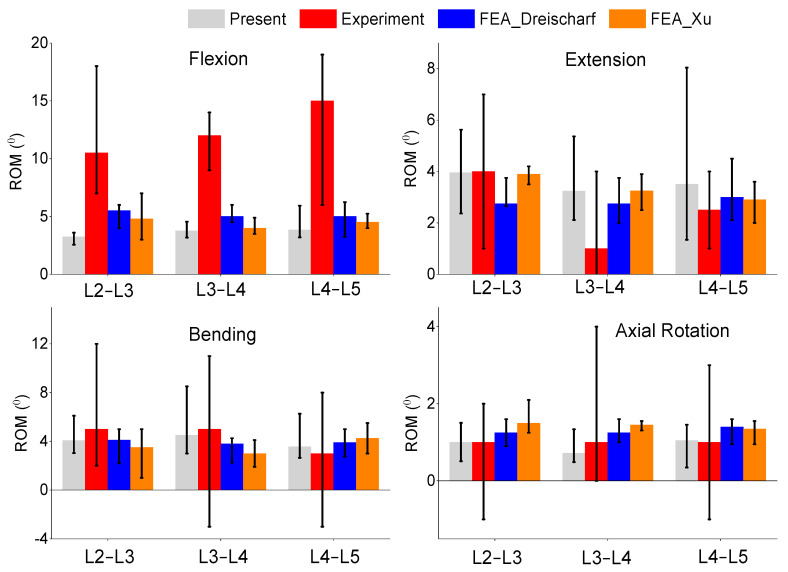
Comparison of median and range of intervertebral rotations of flexion (**top left**), extension (**top right**), lateral bending (**bottom left**), and axial rotation (**bottom right**) from the present study, in vivo experiment results [34], and FEA from the literature [15,25].

**Figure 4 bioengineering-11-00024-f004:**
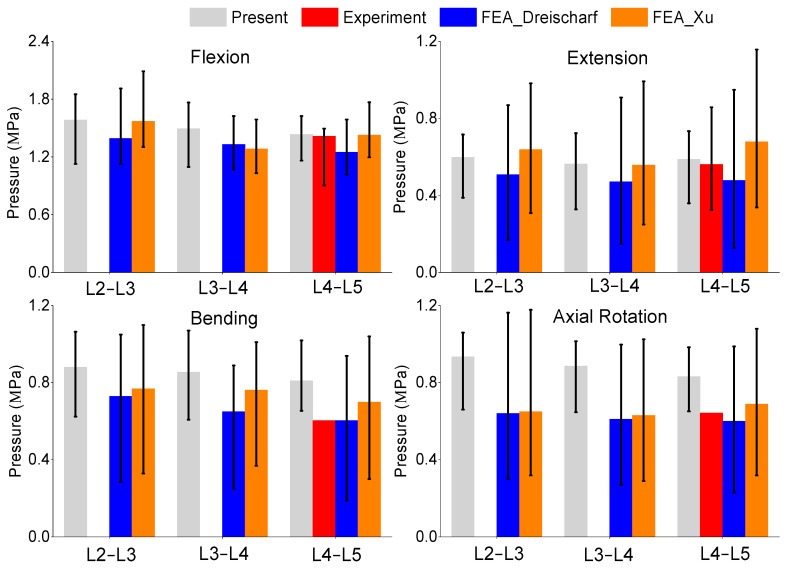
Comparison of median and range of intra-discal pressure of flexion (**top left**), extension (**top right**), lateral bending (**bottom left**), and axial rotation (**bottom right**) from the present study, in vivo experiment results [35,36], and FEA from the literature [15,25].

**Figure 5 bioengineering-11-00024-f005:**
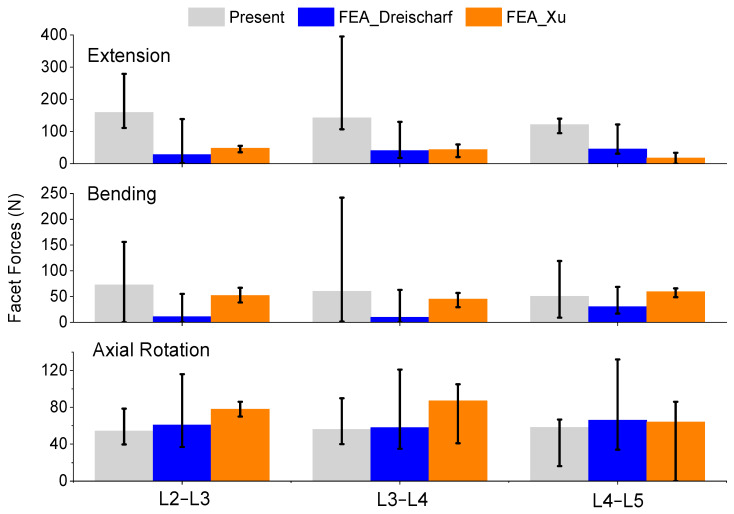
Comparison of median and range of facet forces of extension (**top**), lateral bending (**middle**), and axial rotation (**bottom**) from the present study and FEA in the literature [15,25].

**Figure 6 bioengineering-11-00024-f006:**
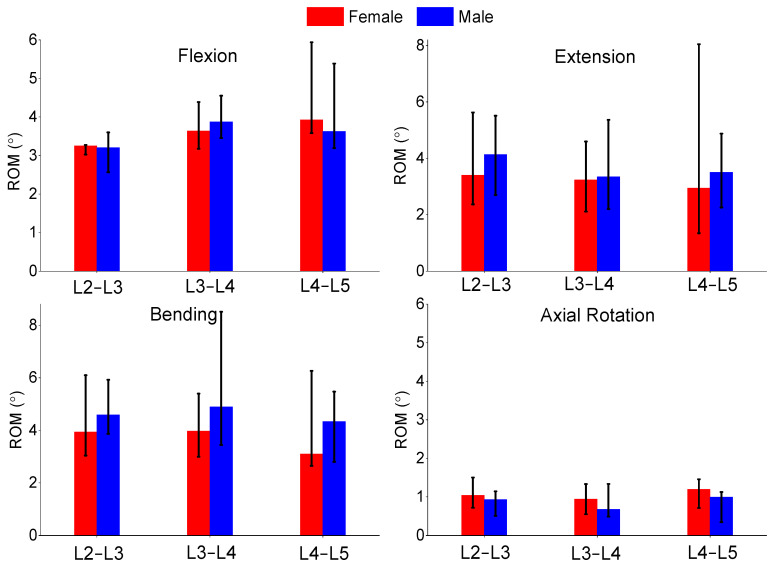
Comparison of median and range of intervertebral rotations of flexion (**top left**), extension (**top right**), lateral bending (**bottom left**), and axial rotation (**bottom right**) for females and males.

**Figure 7 bioengineering-11-00024-f007:**
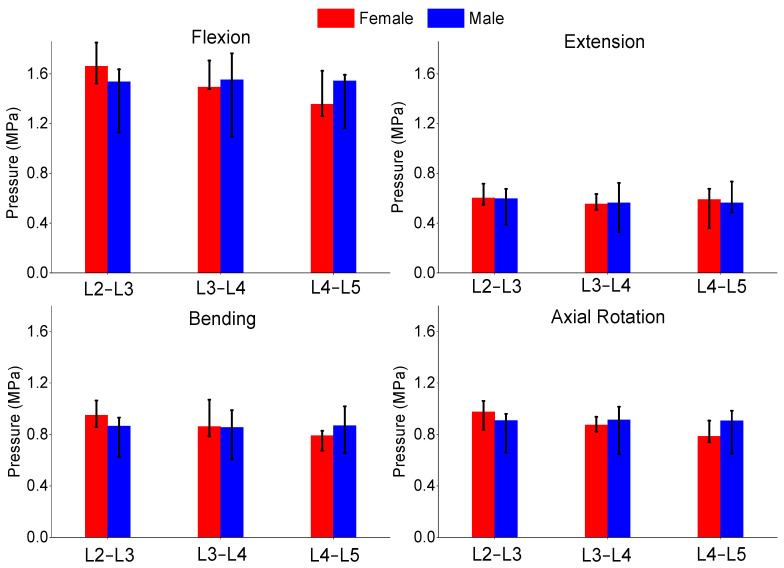
Comparison of median and range of intra-discal pressure of flexion (**top left**), extension (**top right**), lateral bending (**bottom left**), and axial rotation (**bottom right**) for females and males.

**Figure 8 bioengineering-11-00024-f008:**
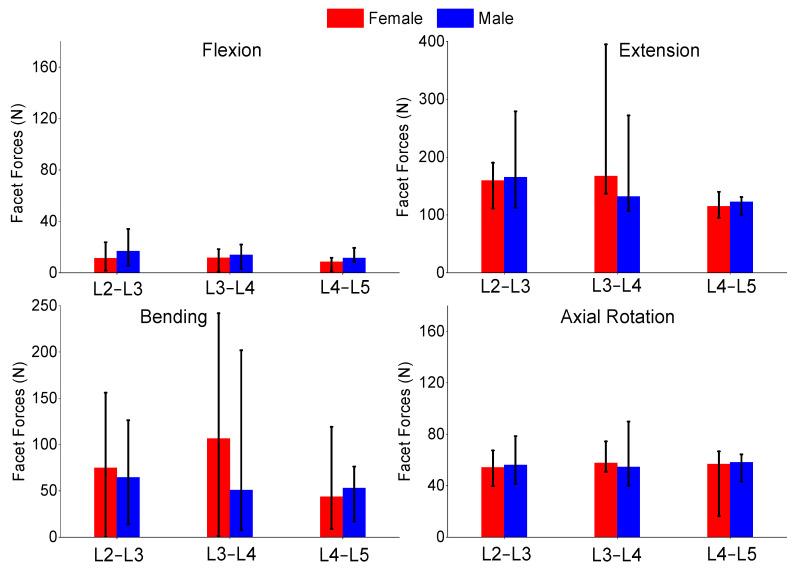
Comparison of median and range of facet forces of flexion (**top-left**), extension (**top-right**), lateral bending (**bottom-left**), and axial rotation (**bottom-right**) for females and males.

**Figure 9 bioengineering-11-00024-f009:**
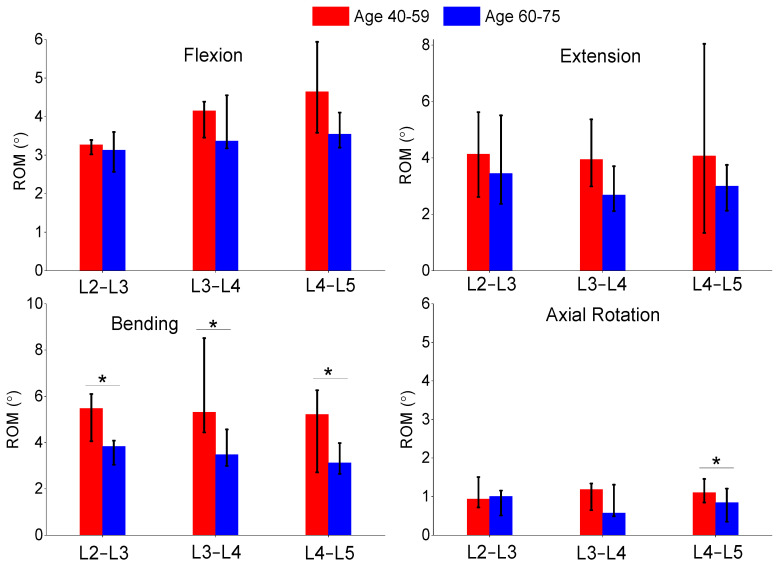
Comparison of median and range of intervertebral rotations of flexion (**top left**), extension (**top right**), lateral bending (**bottom left**), and axial rotation (**bottom right**) for subjects ages 40–59 and ages 60–75. The “*” denotes statistical differences between the groups.

**Figure 10 bioengineering-11-00024-f010:**
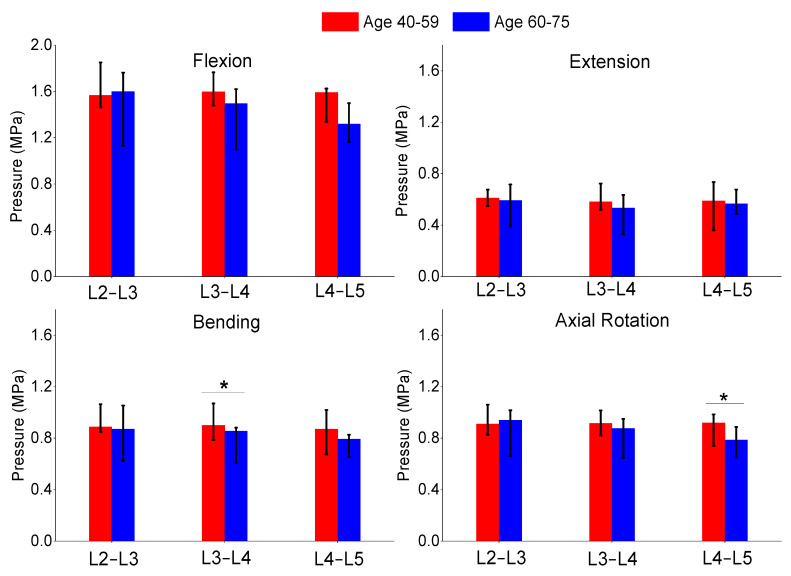
Comparison of median and range of intra-discal pressure of flexion (**top left**), extension (**top right**), lateral bending (**bottom left**), and axial rotation (**bottom right**) for subjects ages 40–59 and ages 60–75. The “*” denotes statistical differences between the groups.

**Figure 11 bioengineering-11-00024-f011:**
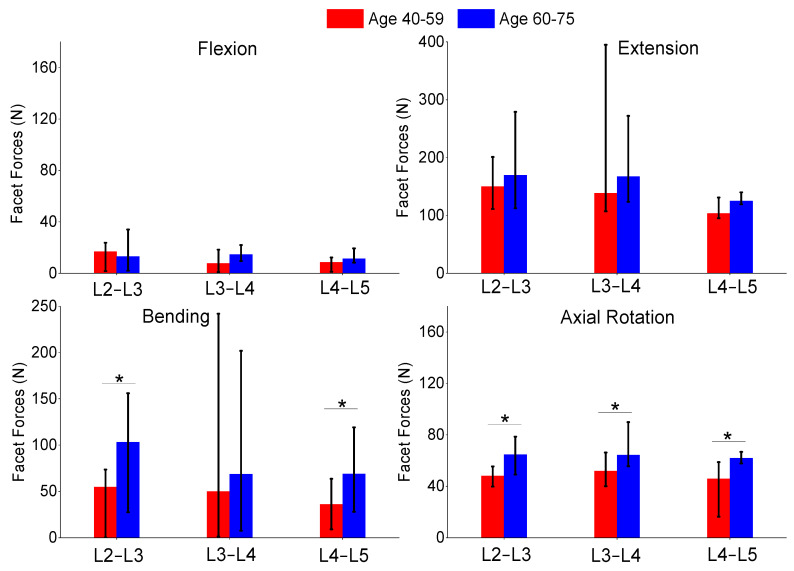
Comparison of median and range of facet forces of flexion (**top-left**), extension (**top-right**), lateral bending (**bottom-left**), and axial rotation (**bottom-right**) for subjects ages 40–59 and ages 60–75. The “*” denotes statistical differences between the groups.

**Table 1 bioengineering-11-00024-t001:** Summary of material boundary conditions.

Component	Material Properties	Reference
Cortical bone	E = 12,000 MPa ν = 0.30	Xu et al. [25]
Posterior elements	E = 3500 MPa ν = 0.30	Xu et al. [25]
Cancellous bone	E = 100 MPa	Xu et al. [25]
Endplates	E = 23.8 MPa ν = 0.40	Xu et al. [25]
Annulus ground substance	Hyperelastic c_1_ = 0.56, c_2_ = 0.14	Schmidt et al. [32]
Nucleus pulpous	Hyperelastic c_1_ = 0.12, c_2_ = 0.09	Schmidt et al. [32]
Ligaments	Nonlinear stress–strain curves, tension only	Averaged values between Eberlein et al. [12] and Shirazi-Adl et al. [33]
ALL	CSA = 35 mm^2^	
PLL	CSA = 15 mm^2^	
FLA	CSA = 75 mm^2^	
FCL	CSA = 50 mm^2^	
TL	CSA = 8 mm^2^	
ISL	CSA = 35 mm^2^	
SSL	CSA = 30 mm^2^	
Annulus fiber	Nonlinear stress–strain curves, tension only	Shirazi-Adl et al. [33]
Layer 1 and 2 (Innermost layer)	Elasticity ratio = 0.65 CSA = 0.20 mm^2^	
Layer 3 and 4	Elasticity ratio = 0.75 CSA = 0.20 mm^2^	
Layer 5 and 6	Elasticity ratio = 0.90 CSA = 0.20 mm^2^	
Layer 7 and 8 (Outermost layer)	Elasticity ratio = 1.00 CSA = 0.20 mm^2^	

**Table 2 bioengineering-11-00024-t002:** Ligament stress–strain curves of the anterior longitudinal ligament (ALL), posterior longitudinal ligament (PLL), transverse ligament (TL), supraspinous ligament (SSL), ligament flavum (FLA), facet capsular ligament (FCL), and interspinous ligament (ISL) (CSA means cross-sectional area). Derived from the works of Eberlein et al. [12].

ALL/PLL/TLL/SSL		FLA		FCL		ISL	
Strain	Stress (MPa)	Strain	Stress (MPa)	Strain	Stress (MPa)	Strain	Stress (MPa)
0	0	0	0	0	0	0	0
0.15	1	0.35	1	0.24	1	0.17	1
0.18	1.5	0.4	1.5	0.28	1.5	0.295	2
0.22	2.5	0.45	2.5	0.315	2.5	0.39	3
0.25	5	0.52	5	0.35	5	0.49	4
0.27	7.5	0.55	7.5	0.38	7.5	0.56	5
0.285	10	0.575	10	0.39	10	0.6	5.6

**Table 3 bioengineering-11-00024-t003:** Tension only fiber stress–strain curves of the innermost layers (1 and 2) to the outermost layers (7 and 8) derived from the works of Shirazi-Adl et al. [33].

	Layers 1 and 2	Layers 3 and 4	Layer 5 and 6	Layer 7 and 8
Strain	Stress (MPa)			
0	0	0	0	0
0.05	29.25	33.75	40.5	45
0.1	47.45	54.75	65.7	73
0.15	57.2	66	79.2	88
0.2	63.05	72.75	87.3	97
0.25	66.3	76.5	91.8	102

## Data Availability

The raw data supporting the conclusion of this article will be made available by the authors, without undue reservation.

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
