# Peer review of "Inter-Specimen Analysis of Diverse Finite Element Models of the Lumbar Spine"

_bioengineering, 2023, doi:10.3390/bioengineering11010024_

Round 1
Reviewer 1 Report
Comments and Suggestions for Authors
1. Strengthen the literature review by citing more recent, relevant articles.
2. Try to add at least two references for the Galerkin Finite Element Analysis..
3. The authors should removeall the typos/grammatical errors from the whole article and improve the language quality in the revised manuscript.
4. More details on the software used is required.
5. Where is the mesh used?
6. What is the novelty of this research work?
7. Comparison and validation of the results may be done with the latest papers published in (2020-2023).
8. Do all terminologies are properly defined in the manuscript. Nomenclature part is required
9. Write the conclusion part in the form of important points.
Comments on the Quality of English LanguageImprove the language quality in the revised manuscript.
Reviewer 2 Report
Comments and Suggestions for Authors
This manuscript described the FEM analysis of lumbar spine (L2-L5) under physiological loading and compared it with the experimental values and previously reported values in the literature.
-At first, it seemed to be of interest, but after going through the experimental and results, I am not certain about the advantage or benefit of the proposed FEM in this study since the simulated results did not provide better results in comparison to the actual measurement or other FEM studies as claimed. Also the use of a larger number of specimens (8 specimens) did not give any benefit as claimed for the study and the variation or the SD were still high as could be seen from the error bars.
- Statistical analysis should be described and added as a subsection detailing the method (either parametric or non-parametric), the significant level used for each result comparison.
-Figs 3-5, it seemed that the FEM analysis from others yielded results consistently close to experimental results and to each other. In contrast, the present study although within the results from other studies, but it differed from experimental and other FEM for certain cases i.e. ROM extension, Axial rotation, etc. Why? This should be discussed on the causes and how to improve.
- It seemed that the referenced values used for analysis and comparison were rather old so the validity of using them is questionable i.e Shirazi-Adl et al., 1986) for input parameters and Pearcy, 1985 for experimental results.
-Error bars were not consistently large or small, but varied depending on the simulated values, location, age. Why is that? What was the cause and the implication?
-Discussion: Should be improved and revised to discuss the results gained from the analysis compared to the actual experiments or other FEM studies whether this study was better or worse and WHY? What is the benefit of the technique in this study and the drawback over other FEM studies? Why age produced different results, not sex? At present, it was quite vague and not into detailed discussion or analysis.
- Conclusion: Should be revised to reflect the concluded remark gained from the study and perspective. At present it was quite a general remark, rther a discussion and some were not supported by the results i.e The methodology in this study will produce an acceptable representation of the lumbar spine (I am not sure to agree with this statement), The inter-specimen analysis indicates that multiple specimens should be used in finite element modeling to account for specimen variability, but the exact number of specimens is still unknown (How to know that it was sufficient?).
- Other points could be found in the attached file. the study and perspective. At present it was quite a general remark, rther a discussion and some were not supported by the results i.e The methodology in this study will produce an acceptable representation of the lumbar spine (I am not sure to agree with this statement), The inter-specimen analysis indicates that multiple specimens should be used in finite element modeling to account for specimen variability, but the exact number of specimens is still unknown (How to know that it was sufficient?).
- Other points could be found in the attached file.

English should be rechecked and revised for better comprehension. At present numerous, unclear or ambiguous sentences can be found in many places, especially in the Results section.
Reviewer 3 Report
Comments and Suggestions for Authors
The manuscript is aimed at very interesting and hot problem i.e. development of the correctly validated FEM model of the lumbar spine. The presented results are also very interesting and looks new. The text is written in a clear and precise manner. However, there are some critical points that should be improved and clarified. Therefore the quality of the current version of the manuscript does not allow me to recommend it for publication.
The main questions and comments are as follows.
1. Subsection “Model Validation” actually describes the verification of the model. However, none of the real parameters and quantities are presented here. Usually, the convergence plots for some integral parameters are presented for this purpose. One of the important parameters for a system like the lumbar spine is its stiffness. More than that, if the authors plan to proceed the developed model for further studies, it would be very interesting to discuss the convergence of the most important local characteristics such as pressure field within the nucleus pulposus.
2. The authors also did not show any fields (distribution) of stress and strain. This is strange for a full 3D model. More of that, in this concern, it is not clear what pressure is plotted in Figs. 4,7,10? Is it the maximum value or the average one? How can we trust the authors that it is the maximum ore the average?
3. In Fig. 3, the results for flexion are 3-4 times less than the experimental data. The authors’s results are also less than those of the other models. This difference should be discussed and explained. Otherwise the conclusion that the model is validated is not correct.
4. Fig. 5, here again we can see the big differences with the other models excepting axial rotation. Perhaps it is important to account for cartilage in the facet joints?
5. In my opinion, the conclusion that “The age of the specimen is more critical than the gender” has not been fully justified. First, I did not see such an effect of age from Figs. 6-11. Second, your model only includes the geometry of the lumbar spine and only for 8 patients. But for the effect of age it is very important to consider the change in mechanical properties of the bone and soft tissues.
6. Please, correct the caption for Fig. 2.
Reviewer 4 Report
Comments and Suggestions for Authors
The article is interesting and well written. the methodological framework of software development and data collection is clear and well written. the results are described in depth and their discussion in line with them and with the state-of-the-art analysis presented in the introduction.
English language is fine. No revisions are to be added.
Author Response
The authors would like to thank the reviewer for the positive and constructive comments on the strength of this study.
Reviewer 5 Report
Comments and Suggestions for Authors
The presented manuscript is very well prepared. The range of results is presented clearly with a discussion in relation to other researchers.
The reviewer only pays attention to such elements as:
- Table 1, CSA size - .20 mm2. The notation is not very understandable. What does the dot before the number mean?
- Line 212-213: What do the authors mean in this notation that the values are within the appropriate range? Here this range needs to be quantified.
These are only minor comments that do not affect the recommendation of the manuscript for further processing. A repeat review is not required.
Author Response
- Table 1, CSA size - .20 mm2. The notation is not very understandable. What does the dot before the number mean?
Response: Thank you for your comment. We have updated the table. CSA means cross-sectional area.
- Line 212-213: What do the authors mean in this notation that the values are within the appropriate range? Here this range needs to be quantified.
Response: Thank you for your comment. We have updated the explanation to “results were combined to be consistent with the literature.”
- These are only minor comments that do not affect the recommendation of the manuscript for further processing. A repeat review is not required.
Response: The authors would like to thank the reviewer for the positive and constructive comments on the strength of this study.
Round 2
Reviewer 1 Report
Comments and Suggestions for Authors
Accept
Comments on the Quality of English LanguageIn general, acceptable.
Author Response
Thank you for your review and comments.
Reviewer 2 Report
Comments and Suggestions for Authors
Authors revised the manuscript according to the recommendation. However, some are still needed to revise further.
- Statistical analysis. Authors revised by inserting sentences in results subsection which is not appropriate. The description of statistical analysis should be add in the materials and methods section as a new subsection since this is a detail of methods used for statistical analysis and applied for all results not just a model validation. In addition, more details should also be included i.e which parametric test/ non-parametric (ANOVA???) tests was used in this study. Any post-hoc test employed?
- The statement of age difference in discussion and conclusion should be carefully refined. Since only 8 specimens/small age ranges were used and only bending and axial rotation were different, not all. These limitations should be noted for readers.
Comments on the Quality of English LanguageEnglish should be rechecked. Several errors and unclear statements could still be found i.e p. 6 Moment and follower preload values depended on the intended motion, p.6 Lateral bending and axial rotation 172 left/right results were combined to be consistent with the literature, p.7 the range was within the 202 FEA literature but was higher than the in vivo measurements, etc.
Reviewer 3 Report
Comments and Suggestions for Authors
1. Again, I must return to Subsection “Model Validation”. Since the authors refused to state that their models are validated, I would suggest to change the name of this subsection. In fact, it verybriefly describes the models verification and the plan of the following study/comparison/analysis.
2. I still do not see that the difference in intradiscal pressure for two age groups is greater than that that for gender. I would recommend correcting this part of the age/gender discussion.
The authors addressed all my comments, but now I have to conclude that the scientific significance of the study is relatively low. The main drawback of the latest version is that the models presented are not yet valid for drawing any serious conclusions.
Round 3
Reviewer 3 Report
Comments and Suggestions for Authors
I thank the authors for their efforts to improve the manuscript.